# Targeting Adrenergic Receptors in Metabolic Therapies for Heart Failure

**DOI:** 10.3390/ijms22115783

**Published:** 2021-05-28

**Authors:** Dianne M. Perez

**Affiliations:** The Lerner Research Institute, The Cleveland Clinic Foundation, 9500 Euclid Ave, Cleveland, OH 44195, USA; perezd@ccf.org; Tel.: +1-216-444-2058

**Keywords:** adrenergic receptor, metabolism, heart, myocyte

## Abstract

The heart has a reduced capacity to generate sufficient energy when failing, resulting in an energy-starved condition with diminished functions. Studies have identified numerous changes in metabolic pathways in the failing heart that result in reduced oxidation of both glucose and fatty acid substrates, defects in mitochondrial functions and oxidative phosphorylation, and inefficient substrate utilization for the ATP that is produced. Recent early-phase clinical studies indicate that inhibitors of fatty acid oxidation and antioxidants that target the mitochondria may improve heart function during failure by increasing compensatory glucose oxidation. Adrenergic receptors (α_1_ and β) are a key sympathetic nervous system regulator that controls cardiac function. β-AR blockers are an established treatment for heart failure and α_1A_-AR agonists have potential therapeutic benefit. Besides regulating inotropy and chronotropy, α_1_- and β-adrenergic receptors also regulate metabolic functions in the heart that underlie many cardiac benefits. This review will highlight recent studies that describe how adrenergic receptor-mediated metabolic pathways may be able to restore cardiac energetics to non-failing levels that may offer promising therapeutic strategies.

## 1. Introduction

Adrenergic receptors (ARs) are members of the G-Protein Coupled Receptor (GPCR) superfamily that bind adrenaline (i.e., epinephrine) and noradrenaline (i.e., norepinephrine) [1]. They regulate the sympathetic nervous system at both central and peripheral sites and modulate the “flight or fight” response to stress. Manifestations of this response include increasing heart rate and contractility, diverting blood flow to essential organs such as skeletal muscle and increasing metabolism to promote survival. ARs are classified into three families: α_1_, α_2_, and β. Each of these families contain three subtypes (α_1A_, α_1B_, α_1D_), (α_2A_, α_2B_, α_2C_), (β_1_, β_2_, and β_3_), and all three families bind epinephrine (Epi) and norepinephrine (NE) with similar affinity. However, they can regulate different physiological processes because they couple to different heterotrimeric and monomeric G-proteins and effector proteins that transduce different signaling pathways (Table 1). AR signaling causes a negative feedback to desensitize and regulate the signal transduction process through activation of GPCR kinases that phosphorylate specific residues on the receptor [2,3].

While β-ARs are well known to be affected during heart failure (HF) and β-AR blockers used in HF treatment, the role of α_1_-ARs, particularly the α_1A_-AR, are recent developments in offering cardioprotective and inotropic benefits during HF. Their roles in regulating metabolic pathways that are altered during HF may be an underappreciated mechanism in their ability to treat HF and is the focus of this review.

## 2. Cardiac ARs

The myocyte expresses both α_1_- and β-AR subtypes. β-ARs are predominately expressed in the myocyte, followed by the β_2_-AR subtype [4,5] in an approximate 80:20 ratio. β-ARs are transduced by the G_s_ G-protein to stimulate adenylate cyclase which produces a transient rise in the second messenger cAMP (Table 1). The main effector of cAMP is protein kinase A (PKA). The major function of β-ARs in the heart are to regulate rate and contractility in response to the sympathetic release of NE and Epi. PKA signaling mediates a positive inotropic and chronotropic response (Table 1) through the phosphorylation of cardiac calcium-handling and myofilaments that control the excitation-contraction coupling such as L-type Ca^2+^ channels, ryanodine receptors, phospholamban and troponin I [6,7,8,9]. The β_2_-AR signals differently from the β_1_-AR in the heart with the β_2_-AR being able to switch coupling to G_i_ (Table 1), resulting in cardioprotective effects against apoptosis [10,11,12] but has little effects on inotropy or chronotropy. There is a small amount (5% of total β-AR pool) of the β_3_-AR in human heart. While the β_1_-ARs regulate positive inotropy, the β_3_-AR is postulated to regulate negative inotropy through its release of nitric oxide [13,14,15,16] (Table 1).

Of the α_1_-ARs, both the α_1A_ and α_1B_-AR but not the α_1D_-AR subtype is expressed in myocytes [17,18,19] (Table 1). The α_1D_-AR is expressed in vascular smooth muscle as are the other α_1_-AR subtypes but is more highly expressed in coronary arteries [20,21,22]. The α_1B_-AR is expressed at higher levels in the heart than the α_1A_-AR based upon radioligand binding studies in rodent and human tissues [18,21,23] but represent a much overall lower density than total β-ARs. The main signaling pathway that is transduced by α_1_-ARs are coupled to the G_q/11_ family of G-proteins. G_q/11_ activates phospholipase Cβ1 (PLCβ1) which cleaves the membrane-bound phosphatidylinositol 4,5-bisphosphate to release inositol triphosphate (IP3) and diacylglycerol (DAG) [24] (Table 1). The IP3 causes a release of intracellular calcium from the endoplasmic reticulum while DAG activates protein kinase C (PKC). α_1_-ARs activate a positive inotropy induced from calcium sensitizing the myofibril filaments [25,26,27] and by phosphorylation of various cardiac proteins by PKC and/or Rho also contributing [28,29,30,31]. 

A main function of all α_1_-ARs is its ability to increase blood pressure via contraction of vascular smooth muscle (Table 1). However, in the heart, studies suggest that the α_1A_-AR subtype mediates positive inotropy [29,32,33,34] while the α_1B_-AR regulates negative inotropy [35,36,37,38,39] which may act as a physiological brake to regulate the robust positive inotropy of β-ARs [40].

While the α_2_-ARs are not present in the myocyte, the α_2B_-AR subtype does control vascular contraction and blood flow [41] while the α_2A_-AR and α_2C_-AR subtypes are located on sympathetic nerve terminals to regulate the release of NE [42] (Table 1) and thus, can still influence the onset and progression of HF [43]. During failure, there is a chronic release of catecholamines from the sympathetic nerves and a genetic dysfunctional variant of the α_2C_-AR has been described in a human heart failure patient linked with poor prognosis [44].

## 3. ARs in Heart Failure

In HF, the sympathetic nervous system tries to compensate from the loss of contractility by causing high levels of catecholamine release, referred to as overdrive. The elevated plasma levels of NE cause β_1_-ARs to be desensitized and downregulated by as much as 50% with β_2_-ARs levels remaining unchanged as shown by radioligand binding studies [4,45]. Chronic stimulation of the β_1_-ARs, while beneficial in the short-term for HF by increasing inotropy, long-term signaling causes necrosis and cell death due to cAMP-mediated calcium overload, so β_1_-AR agonists are not a viable therapy [46,47,48,49]. In contrast, β_2_-AR stimulation is thought to be cardioprotective, protecting against apoptosis [11,50,51] and may improve function [51,52], but becomes more diffuse and non-compartmentalized in the myocyte during HF and may not signal properly [53]. β-AR blockers have long been used as a standard of treatment for HF [45,54,55,56,57]. Besides a direct blocking action of catecholamines, β-AR blockers also can re-sensitize the β-ARs preventing their downregulation and repopulating the myocyte cell surface [58].

β_3_-AR stimulation has been proposed as a brake system for β_1_- and β_2_-AR overstimulation during HF since they may couple to G_i_ in the myocardium [13,14]. Besides having its own cardioprotective benefits [59], the β_3_-AR lacks GPCR kinase recognition sites [60] and so will not become desensitized as the other β-ARs during sympathetic overdrive during HF. However, the potential use of β_3_-AR agonists or antagonists in HF is still highly debated.

Changes in α_1_-AR receptor protein levels during HF are also controversial with receptor levels being either unchanged [21,61], increased [17,62,63,64] or decreased [65,66,67]. The vast variability in these studies of changes in α_1_-AR protein levels could be due to the severity, or degree of HF [66], the amount of sympathetic overdrive and plasma levels of catecholamines [68] or the etiology of HF (ischemic versus non-ischemic) since α_1_-ARs are known to increase receptor density during ischemia or low oxygen conditions [69,70,71,72,73,74]. 

Early assessments of α_1_-AR effects on the heart were first reported as secondary and minor to β-ARs [1] but with better tools available, the α_1A_-AR has been shown to have a robust positive inotropic response in the human heart and remains functional during failure [34,37,75,76,77]. It is believed that α_1A_-AR stimulation is adaptive in the heart and stimulation of this subtype may provide therapeutic benefit during failure [1,34]. Transgenic and knock-out (KO) mouse models indicate that mice are protected against HF [78] or myocardial infarction [79]. The α_1A_-AR selective agonist, A61603 or dabuzalgron can increase survival and prevent cardiac damage [80,81]. In contrast, stimulation of the α_1B_-AR subtype is maladaptive in the heart and can lead to dilated cardiomyopathy or dysfunction [82,83,84], progressing to HF [85]. During HF and β-AR downregulation, the α_1A_-AR’s contractile effects in the myocyte may become important in mitigating the loss of β-AR-mediated inotropy [37,38,39].

## 4. Metabolic Regulation of Normal vs. Failing Heart

The heart has the highest metabolic rate and energy requirement of all organs other than the kidney [86] and HF has often been characterized as “an engine out of fuel” [87]. To satisfy these high energy demands, the normal heart has a high degree of flexibility to metabolize various substrates to generate adenosine triphosphate (ATP) such as utilizing glucose, lactate, fatty acids, and ketone bodies. This process is tightly regulated but has a high degree of plasticity and interdependence between the substrates utilized. The pathology of HF causes alterations in the heart’s ability to regulate and metabolize these substrates resulting in energy deficits that contribute to aversive outcomes and poor prognosis [88,89,90].

### 4.1. Glucose Utilization

Various carbohydrates such as glucose, glycogen, and lactate are used by the heart to produce ATP. Glucose is taken up by the myocyte through glucose transporters (GLUTs) GLUT1 and GLUT4 [91], then metabolized by oxidation to produce ATP, CO_2_, and H_2_O. This process first entails glycolysis which does not require oxygen (i.e., anaerobic) and occurs in the cytosol. Phosphofructokinase 1 is the step that commits glucose to undergoing glycolysis [92]. If sufficient oxygen (i.e., aerobic) is present, pyruvate then enters the mitochondria and is metabolized further by the citric acid cycle or Krebs cycle and the electron transport system, producing in total 30 ATP molecules per molecule of glucose. If the heart is ischemic (i.e., lack or low oxygen levels), only 5 ATP molecules are produced, a vast difference. The oxidation of glucose is mainly regulated by the pyruvate dehydrogenase (PDH) complex. A schematic of glycolysis and glucose oxidation in the myocyte is shown in Figure 1.

An alternative pathway for glucose utilization is the pentose phosphate pathway (PPP) which is the main generator of nicotinamide adenine dinucleotide phosphate (NADPH) in the cytoplasm [93]. It also generates ribose sugars which are used for nucleotide synthesis. The first step of the PPP is the oxidation of glucose-6-phosphate (produced through glycolysis) which uses the rate-limiting enzyme, glucose-6-phosphate dehydrogenase (G6PD) to generate NADPH (Figure 1). NADPH is required by several defensive redox antioxidant pathways for survival of the cell against reactive oxygen species (ROS), particularly in organs with high energy requirements, such as the heart [94]. In a normal heart, approximately 2–4% of oxygen used during metabolism during oxidative phosphorylation is converted to ROS [95,96,97]. In a healthy cell, the rate of superoxide production is kept in check by its dismutation to water and oxygen via the enzyme superoxide dismutase to hydrogen peroxide, then eventually to water [93,98]. An imbalance in this system can occur due to excessive ROS production or insufficient antioxidant breakdown. Increased ROS species have been well documented to occur in HF patients [99,100,101,102,103,104].

A loss of that protection from ROS damage in the heart can result from deficiency in G6PD and shown to occur during ischemia [105], and HF [105,106,107,108]. Unloading of the failing heart using a left ventricular assist device increased the metabolic flux into the PPP [108,109]. However, persistent activation of this pathway or upregulation of G6PD can occur during severe HF when glucose oxidation becomes severely limited and forces more of the glucose to enter the alternate PPP [110,111]. In this case, the PPP now promotes superoxide generation and oxidative stress in the failing heart due to increases in NADPH oxidase [112,113]. Increases in G6PD in the heart can also occur in diabetic or obese patients before cardiac pathology develops leading to more adverse outcomes [114]. The role of G6PD is even more complicated because the same study that associated deficiency and loss of ROS damage protection during HF [106] also found evidence of improvement in coronary artery disease and has been confirmed in additional studies [115,116]. Several studies on the metabolomic profiles of HF patients indicated a large array of metabolic dysfunction including the PPP [90,91,117,118]. The regulation of the PPP may be more important than previously realized in HF because decreased G6PD levels are the most common enzyme deficiency in humans [119,120]. The genetic mutation in G6PD that causes decreased expression is X-linked recessive [121], hence it is most commonly present in males and could be a possible explanation of why men developed HF more frequently than women.

### 4.2. Fatty Acid Metabolism

In the normal adult heart, about 95% of its ATP requirements are due to mitochondrial oxidative phosphorylation with 60–80% of that from fatty acid oxidation [122,123,124,125,126]. Fatty acids can freely cross the cell membrane or are transported and activated in the heart via transport proteins (fatty acid transport protein, FATP) and are activated by conversion to long-chain fatty acyl-CoA esters. They are then exported into the mitochondria for β-oxidation [127]. While medium chain fatty acids can cross freely into the mitochondria, a transport mechanism called the carnitine shuttle is required to shuttle long-chain fatty acids into the mitochondria using the enzymes carnitine palmitoyl transferase 1 and 2 (CPT1 and 2) [128,129]. Once in the mitochondria, β-oxidation begins and is cyclic, shortening the fatty acyl-CoA esters by 2 carbon units each cycle till the chain is fully broken down, liberating acetyl-CoA (which enters the TCA cycle for further metabolism), and producing nicotinamide adenine dinucleotide (NADH) and 1,5-dihydro-flavin adenine dinucleotide (FADH_2_) which are oxidized in the electron transport chain (i.e., oxidative phosphorylation) to produce ATP and water. A schematic of fatty acid oxidation in the cardiac myocyte is shown in Figure 2.

As the heart fails, its ATP levels drop by 30–40% during the advanced stages of failure compared to the normal heart [130,131,132]. The drop in ATP production during HF is due to several factors such as increased ROS, altered calcium handling, defects in the consumption rates of oxygen, mitochondrial dysfunction, decreased electron transport chain activity, which all have effects on ATP generation during oxidative phosphorylation [87,89,126,133,134,135,136]. This energy deficit during HF is responsible for the loss of contractile function as contraction alone requires more than 60% of the available normal ATP pool [126]. The little ATP that is left after contraction during failure results in insufficient energy to regulate the other critical functions of the heart such as calcium handling, cellular homeostasis, and maintaining membrane potentials resulting in decrease mitochondrial turnover and biogenesis [137,138,139,140,141].

The metabolic profiling of induced-HF in various animal models does show considerable variability and the need to be cautious in interpretations of the early metabolic changes. During pressure overload-induced HF, there is a decrease in both fatty acid and glucose oxidation [142,143]. In other animal models of HF, such as rapid ventricular pacing, there are decreased rates of fatty acid oxidation but increased rates of glucose oxidation [110,143,144,145]. These differences might be due to the severity of the HF, as metabolic flexibility in substrate use that occurs in the normal heart is lost during the progression to severe HF, resulting in decreased ability to increase either fatty acid or glucose uptake [146,147]. 

Another explanation for the metabolic disparity in the various types of HF models may be dependent upon whether there is failure with preserved ejection fraction (HFpEF) or with reduced ejection fraction (HFrEF) [148,149,150]. Volume or pressure overload HF models may not cause a reduction in ejection fraction. HFpEF accounts for almost half of the HF population but has similar morbidity and mortality rates as HFrEF [137,151,152,153]. In HFrEF models, the metabolic changes mostly include a decrease in both fatty acid and glucose oxidation but increases in glycolysis [144,146,149,154,155,156]. In contrast, in HFpEF models, fatty acid oxidation and glycolysis are increased but glucose oxidation is decreased [157,158,159,160,161,162,163]. As both HFrEF and HFpEF have in common a decrease in glucose oxidation resulting in cardiac energy inefficiency, metabolic treatments that can increase glucose oxidation may be the most universal in treating HF.

Abnormally high rates of glucose metabolism are one of the earliest metabolic changes that occurs during the development of HF [154,164]. The shift to increased utilization of glucose as opposed to fatty acids is postulated to occur because it spares the use of oxygen [125,165]. While glucose uptake is generally preserved or increased during failure, there is a metabolic shift from oxidative pathways to glycolysis and an uncoupling between glycolysis and its complete oxidation. This is because of reduced capacity to enter the TCA cycle [166] or oxidative phosphorylation due to defects in electron transport [104,167,168,169,170], resulting in energy inefficiency and functional weakening of the heart [171,172]. An increase in anaerobic glycolysis uncouples the oxidative process and results in the generation of lactate and protons [173,174,175]. The resulting decrease in pH reduces the contractility of the heart [176,177] and decreases cardiac efficiency [125,161,178,179]. Hence, therapeutics that can optimize glucose utilization and oxidation may reduce the severity of HF by increasing energy efficiency of glucose [179,180,181,182,183,184].

### 4.3. General Metabolic Therapies for HF

Stimulating glucose uptake and oxidation in the heart has been shown to improve the rate of recovery from ischemic damage [179,182,185,186]. This can occur through increased GLUT expression or translocation, increased activity of phosphofructokinase 1 or flux into glycolysis [109,110]. However, current thinking is that increasing glucose uptake without also increasing glucose oxidation may cause an uncoupling of the pathways and lead to cardiac inefficiency [125]. Activation of pyruvate dehydrogenase (PDH), a rate-limiting enzyme in glucose oxidation, can increase glucose oxidation rates. PDK kinase inhibitors such as dichloroacetate, can stimulate PDH activity improves energy efficiency and reduces oxygen consumption to improve function during HF [187] but has a short half-life and would need to be continuously infused.

Inhibition of fatty acid oxidation causes a compensatory increase in glucose oxidation and improves cardiac metabolic efficiency, decreases mitochondrial oxidative capacity, and has also been used for a number of therapeutic strategies in HF [156]. Carnitine palmitoyltransferase-1 (CPT1) is an enzyme involved in the uptake of fatty acids into the mitochondria [188]. Inhibitors of CPT1 decrease fatty acid oxidation and increase glucose oxidation in the heart [180,188,189] and can improve outcomes in HF [190,191] but can lead to serious side effects and high doses cause impaired energetics [192,193]. Inhibitors of the last enzyme in fatty acid oxidation, long-chain 3-ketoacyl CoA thiolase, are showing a better therapeutic profile in treating HF [194,195,196]. 

Since fatty acid oxidation provides much greater ATP production compared with glucose, therapeutic treatments that revert the heart back to using fatty acids as substrates might also be an option for treating HF, but several models of HF have shown downregulated or have dysfunctional enzymes or proteins involved in the fatty acid oxidation process [142,197,198,199]. If these defects can be reversed through mechanical unloading or other means, targeting peroxisome proliferator-activated receptors (PPARs) or mitochondrial fatty acid metabolizing proteins could provide therapeutic treatments for HF. 

## 5. ARs in Cardiac Metabolism and as Potential Therapeutics

ARs regulate many aspects of metabolism in the substrate preference, mobilization, and utilization as it relates to normal heart function and changes that can occur during HF [90,117,200,201]. Mouse embryos with dopamine β-hydroxylase gene KO which prevents the biosynthesis of NE or Epi, have decreased glucose metabolism, oxygen consumption, ATP levels with elevated concentrations of ADP, leading to HF and death [90,200,201]. These deficiencies could be reversed by administration of either the β-AR agonist, isoproterenol, or the α_1_-AR agonist, phenylephrine [201]. The addition of pyruvate also led to a recovery of the ATP loss, suggesting that the loss of cardiac glycolysis prevented sufficient substrate for aerobic respiration [90]. Targeted disruption of the pathway that leads to only Epi biosynthesis does not affect heart development nor develops HF [202,203]. These results suggest that NE by stimulating both α_1_- and β-ARs are essential for heart development and metabolism by regulating the embryonic shift from anaerobic glycolysis to aerobic metabolism and oxidative phosphorylation in the heart.

In general, catecholamines have also been shown to increase glucose uptake, glycolysis, and glucose and fatty acid oxidation in the adult heart [204,205,206,207] driving fatty acid oxidation at much less levels (10% increase) versus a much larger proportion (410% increase) in glucose oxidation [208]. Catecholamines also increase lipolysis in fat cells, reduce insulin release and insulin sensitivity which can lead to metabolic effects on the heart [209,210,211]. 

### 5.1. α_1_-ARs

Early studies on the role of α_1_-ARs in metabolism have shown increased gluconeogenesis and ketogenesis in both the liver [212,213,214,215,216] and in the kidneys [217,218]. The mechanism is through calcium release and the phosphorylation of glycogen phosphorylase [219,220]. Both the liver and kidneys are gluconeogenic organs and both can increase systemic glucose production. Glucose and sodium metabolism are linked as the kidneys reabsorb glucose through sodium-glucose cotransporters (SGLT) 1 and 2. SGLT inhibitors and inhibition of glucose reabsorption can improve hemodynamics to reduce adverse outcomes in HF [221,222], postulated to occur through increasing fuel efficiency, utilization, and oxygen delivery [221] SGLT inhibitors also induce a fast-like metabolic state that enhances gluconeogenesis and ketogenesis [223], unlike traditional antihyperglycemic agents such as insulin or metformin which suppresses both gluconeogenesis and ketogenesis [224,225]. Ketone bodies are highly efficient source of energy and improve heart work efficiency [226] and more so during HF in the diabetic where glucose metabolism is impaired due to insulin resistance [227]. Ketones are also anti-inflammatory [228,229,230] and antioxidant [231], all protective mechanisms in HF. We speculate that α_1_-AR agonists may confer part of their cardioprotective benefits and limit the progression of HF, particularly in the diabetic heart, by metabolically increasing glucose and ketone body availability through enhanced glucogenesis and ketogenesis. 

Glucose uptake is increased in the heart during ischemia via translocation of the glucose transporters GLUT 1 and GLUT 4 [232]. Both GLUT isoforms 1 and 4 are decreased in human HF [111]. α_1_-AR stimulation can increase glucose substrate availability by increasing its uptake into the heart or in myocytes [232,233,234,235,236] and also in a wide variety of cell lines and cell types such as L6 and C2C12 muscle cells [237,238,239], and adipocytes [240,241,242,243,244]. The mechanism of glucose uptake in the heart is through α_1_-AR stimulation of PKC activation which increase translocation of GLUT 1 or 4 to the membrane resulting in GLUT activation [234,245] (Figure 1). α_1_-AR mediated ^3^H-deoxyglucose uptake was blocked by an inhibitor of GLUT 1 or 4 translocation and by the PKC inhibitor rottlerin or siRNA against PKCδ [246]. α_1A_- and α_1B_-AR KO mice [23,246] or transgenic mice designed with large fragments of their endogenous promoters to drive systemic overexpression of the α_1A_- or α_1B_-AR subtypes with constitutively active mutations (CAMs) [247,248,249] were used to discern effects of the specific α_1_- AR subtypes on metabolism. The α_1A_-AR and not the α_1B_-AR subtype appears selective for glucose uptake in the heart as only transgenic CAM α_1A_- but not CAM α_1B_-ARs increased glucose uptake in the heart and plasma membrane translocation of both GLUT1 and GLUT4. In congruence, knockout (KO) of the α_1A_-AR but not α_1B_-AR mice decreased glucose uptake and GLUT translocation [234,250]. The α_1A_-selective agonist, A61603 also increased glucose uptake into primary myocytes [235]. α_1A_-AR stimulation also increased glucose uptake during glucose-starved conditions of ischemia and protected against annexin V^+^ apoptosis and increased levels of lactate dehydrogenase [232,250]. These results suggest that α_1A_-AR-mediated regulation of glucose uptake in the heart would provide the needed increased metabolic requirements in HF but also provide cardioprotective benefits against apoptosis and cell death. 

Without a concomitant increase in glucose oxidation, an increase in glucose substrate is not energy efficient and of limited benefit in HF. Recently, studies have indicated that α_1A_-AR agonists would also not only increase glucose uptake and availability but also increase its oxidation in the heart. Using radioactive tracers of ^14^C-glucose or ^14^C-palmitate to measure oxidation rates in primary cardiac myocytes, α_1_-AR stimulation with phenylephrine increased glucose oxidation which was blocked by an inhibitor to AMP-activated protein kinase (AMPK) [236], under both normal and diseased (i.e., ischemic) conditions. Therefore, α_1A_-AR agonist treatment may promote energy efficiency and increase the needed ATP in the failing heart by increasing both glucose substrate availability and its subsequent oxidation.

AMPK is a key regulator of energy metabolism and plays a cardioprotective role. AMPK activation is protective in the heart against ischemic damage [251,252,253], cardiac hypertrophy [254] and decreases inflammation and fibrosis [255,256,257,258]. AMPK is an energy sensor that measures the ratio of AMP to ATP to regulate ATP-generating pathways but also prevent oxidative stress by improving and NAD^+^ homeostasis [259,260,261]. AMPK regulates glucose metabolism in myocytes by increasing the translocation of GLUT and activating PKC to increase glycolysis during ischemia [262,263] (Figure 1). α_1_-ARs have been readily shown to activate AMPK in the heart [253,255,264,265]. α_1_-AR mediated ischemic protection is mediated through PKC and AMPK [253] and induces GLUT4 upregulation [251,266] and is shared with the α_1_-AR signals that mediates glucose oxidation [236,250] (Figure 1). Therefore, α_1_-AR metabolic effects and protective signals in the heart may both be regulated through AMPK.

While the systemically expressing CAM α_1_-AR mice can improve glucose tolerance and the α_1A_-AR subtype specifically regulated glucose uptake in the heart, whole body indirect calorimetry found that both CAM α_1A_ and CAM α_1B_ mice had increased whole body fatty acid oxidation by increasing its preference to burn fatty acids as an energy substrate even though they were fed a normal chow diet [234]. In congruence, α_1_-AR KO mice from both subtypes increased whole body glucose oxidation by increasing the preference to burn carbohydrates [234]. The ability to regulate both glucose and fatty acid oxidation suggests that α_1_-ARs may regulate and switch energy fuel preference through coupling to AMPK. α_1_-ARs can regulate AMPK activity not only in the heart [253,257,264,265] but in a wide variety of cell types such as in liver [267], adipose [268], skeletal muscle [239,269] and CHO transfected cells [270]. It is likely that α_1_-AR CAM mice increased whole body fatty acid oxidation through stimulating fatty acid oxidation in the skeletal muscle as that muscle utilizes 40–50% of a body’s whole energy metabolism. The CAM α_1_-AR mice also had increased levels of leptin in the plasma while α_1_-AR KO mice had decreased leptin levels [234]. Leptin can increase the oxidation of glucose in skeletal muscle in the absence of insulin [271,272] or can stimulate fatty acid oxidation in skeletal muscle by activating AMPK and inhibiting acetyl coenzyme A carboxylase, which inhibits carnitine palmitoyltransferase 1 through decreased levels of malonyl-CoA [269] (Figure 1 and Figure 2). During HF where there is metabolic inflexibility on the use of fuel substrates [273,274], the ability of α_1_-AR activation to regulate and shift energy pathways as needed may lead to better outcomes. 

α_1_-AR stimulation can also activate PPARs, in addition to AMPK, to regulate oxidative phosphorylation, glucose homeostasis, ROS, and hypertrophic responses in myocytes [274,275,276,277,278]. PPARs are crucial to maintain normal cardiac function, its energy requirements, and regulates many key metabolic oxidative processes and mitochondrial biogenesis and function [279,280,281]. PPARs are also cardioprotective, particularly PPAR [282]. Cardiac-targeted KO of PPAR decreases basal fatty acid oxidation leading to cardiac dysfunction, lipid accumulation and HF [283]. Cardiac-targeted overexpression of PPARβ/increased glucose utilization [281]. Both PPARα and PPAR activation as well as α_1_-AR agonists increased AMPK phosphorylation and glucose uptake in the heart [274]. α_1_-AR stimulation in the heart reverses ROS and mitochondrial dysfunction when co-treated with PPARα agonists [275,276] or PPAR co-activators [277] (Figure 1). As PPAR activators can rescue HF through metabolic alterations and are being pursued as therapeutics [284,285], PPAR activators have different and sometimes detrimental outcomes in noncardiac tissues and may induce tumorigenesis [286,287,288,289]. Therefore, α_1_-AR agonists may offer a better treatment option in HF by its ability to target PPAR signals in the heart. 

α_1_-ARs, and the α_1A_-AR in particular, also have general effects on glucose homeostasis that may be of benefit to treat HF. α_1A_- and α_1B_-AR KO mice [23,246] or the CAM transgenic mice [247,248,249] were assessed for whole body metabolic changes. Both CAM α_1A_ and CAM α_1B_-AR transgenic mice had an increased tolerance for glucose. However, CAM α_1A_-AR mice were more robust in glucose control and only the CAM α_1A_-AR mice had statistically lowered fasting glucose levels. The α_1A_-AR KO mice had elevated blood glucose after fasting [234]. The CAM α_1A_-AR mice also specifically reduced fasting plasma triglycerides levels while only the α_1A_-AR KO mice had elevated levels [250]. α_1_-AR agonists have also been shown to reduce serum triglycerides [290]. In a metabolomic analysis, the α_1A_-AR selective agonist, A61603, produced a reduction in polyunsaturated fatty acids in the heart [291]. While there is a report that the α_1B_-AR KO mice had impaired glucose homeostasis, insulin resistance and reduced glycogen synthesis [292], the phenotype only appeared when the mice were fed a high fat diet. These results suggest that an α_1A_-AR agonist may provide improved blood lipid and carbohydrate profiles if used as a potential therapeutic to treat HF.

Additional beneficial effects on glucose utilization were shown when either α_1_- and β-AR stimulation increased PPP activity in the heart through increasing the mRNA and protein synthesis of G6PD [293,294,295,296] (Figure 1). In corroboration, carvedilol which antagonizes both the β- and α_1_-ARs [297], blocks PPP activity [298]. Catecholamine deletion due to dopamine β-hydroxylase KO also produced decreased G6PD activity [91]. Therefore, an additional benefit of α_1A_-AR agonism to treat HF would be its generation of NADPH through the PPP to protect the heart against ROS damage.

### 5.2. α_1A_-AR Therapeutics

The above review suggests that the α_1A_-AR subtype may be a therapeutic target for its ability to metabolically protect the heart during failure. However, α_1A_-AR agonists have not been extensively developed in the past because of its ability to also increase blood pressure [299,300,301,302,303]. However, to circumvent non-desirable blood pressure effects, α_1A_-AR agonists are being currently developed that either bias or allosterically prevent signals (i.e., IP3 or calcium release) away from those that induce the blood pressure response. 

α_2_-AR and not α_1_-AR agonists commonly contain the imidazoline pharmacophore and, in general, have better selectivity for α_2_-ARs. α_2_-AR agonists reduce blood pressure by decreasing NE release at the α_2A_-AR autoreceptor on sympathetic nerve endings [304] and also are weak antagonists at the α_1_-AR [305]. However, depending upon the substituents off the imidazoline ring structure, some imidazolines can be designed to become high affinity α_1_-AR selective agonists [306,307,308]. Imidazolines also have about 50-fold higher selectivity for the α_1A_-AR subtype compared to the α_1B_- or α_1D_-AR subtypes [309,310]. There are several imidazolines (i.e., cirazoline, A61603, dabuzalgron) that are α_1A_-AR selective and shown to reduce stress urinary incontinence without a strong response on blood pressure by biasing the signaling towards cAMP and not the IP3/Ca^+2^ response [311,312,313,314] and demonstrated to have improve function or protect during HF [38,39,80,81]. The ability of α_1A_-AR structured imidazolines to separate cardioprotection from the blood pressure effect would depend upon the therapeutic index or the dose of the drug that can separate therapeutic efficacy from toxic side effects. Dabuzalgron (aka Ro 115,1240), for example, was taken to Phase ll clinical trials showing improvement in stress urinary incontinence with little or no cardiovascular effect [311] but did not meet the efficacy hurdle as stated in the clinical trial [315].

Another area of drug development for α_1A_-AR agonists are positive allosteric modulators (PAMs) that increase a receptor’s function but does not bind to the orthosteric (i.e., endogenous) site that agonists bind, such as NE [316]. Allosteric modulators result in greater selectivity by binding to non-conserved regions of the receptor and several advantages over orthosteric agonists, such as usually having conformational bias that can alter the receptor’s signaling pathways [316]. There are now many GPCR allosteric modulators in clinical trials [317,318]. The first PAM at the α_1_-ARs with selectivity for the α_1A_-AR subtype has been developed [319] that can improve the cognitive functions in an Alzheimer’s Disease mouse model without increased blood pressure. 

### 5.3. β_2_-ARs

Similar to α_1_-ARs, β_1_-AR but not β_2_-AR stimulation increases glycogenolysis by the phosphorylation and activation of glycogen phosphorylase [320]. Under increased workload and higher energy requirements in the heart during failure, β_1_-AR agonists can increase anaerobic metabolism when needed [321] and may do so through increased glycogenolysis and subsequent glucose metabolism [208,322,323]. While β-AR stimulation of cAMP increases glucose uptake via GLUT translocation in skeletal cells [324,325], but it has not been shown in myocytes directly. β-AR agonism can reduced the levels of metabolites found in the Krebs cycle, glycogen metabolism, and glycolysis [326], suggesting that β-AR stimulation increases the oxidation of glucose in the heart and may benefit HF. β_1_-ARs appear to increase oxidative metabolism at a greater rate than the β_2_-AR [327] and may explain why downregulation of the β_1_-AR in HF is detrimental. While inotropic interventions for HF might improve short-term hemodynamics, β-AR agonists have long-term effects that have failed to improve outcomes and can even worsen as excessive release of catecholamines induces apoptosis, increased heart rate, and arrhythmias [328,329]. 

β_3_-ARs have been mostly studied in adipose cells to increase cAMP levels, activation of lipase, and thermogenesis [330,331]. Most studies suggest that the β_3_-AR produces a negative inotropy in opposition to positive inotropy regulated by the β_1_- or β_2_-AR [14,16], is cardioprotective [59], and upregulated during human HF [60,332]. As the β_3_-AR does not desensitize and may even increase during HF [65,333,334,335], a β_3_-AR agonist may also provide metabolic benefit. However, the role of the β_3_-AR in HF is controversial. One study suggests that cardiac function can be recovered in HF by blocking the β_3_-AR which improves the energy efficiency in myocardial tissues by suppressing iNOS expression [336]. A β_3_-AR KO mouse model is cardioprotective [337]. In contrast, there are far more abundant studies that indicate that β_3_-AR stimulation or cardiac-overexpression is cardioprotective in many models of ischemia, pressure overload and hypertrophy through inducing nitric oxide (NO) [59,338,339,340,341,342,343,344]. Therefore, β_3_-AR agonism may still be a viable metabolic therapeutic candidate for HF.

The β_3_-AR agonist, mirabegron, can improve glucose homeostasis in insulin-resistant obese humans although this occurs through its function on brown adipose tissue [345]. Mirabegron reduced adipose tissue dysfunction and resulted in improved whole-body glucose tolerance, increased lipolysis and expression of PPAR and enhanced oxidative capacity [345,346]. Mirabegron and other β_3_-AR agonists improved glucose tolerance, utilization [347] and insulin sensitivity in lean mice [348] or mice fed a high fat diet [349,350]. β_3_-AR agonism is anti-atherosclerotic in apoE KO mice with decreased plasma triglycerides, low density lipoproteins, cholesterol, and increased insulin sensitivity and PPARα and expression in the liver [351,352]. These results suggest that β_3_-AR agonists can regulate whole body lipid and glucose metabolism. While a clinical trial of mirabegron failed to improve left ventricular ejection fraction in HF [353], there was significant improvement in a subset of patents with ejection fractions under 40%. This suggest that mirabegron may be therapeutic in HF patients with reduced ejection fraction. There is an ongoing large-scale clinical trial to confirm this effect of mirabegron therapy on improving function in patients with progressive left ventricular remodeling [354]. 

### 5.4. β_1_-AR Therapeutics

With the subsequent downregulation of β_1_-ARs in HF and worse outcomes with long-term stimulation, the metabolic effects of β-blockers to treat HF are preferred. β-AR blockers are a standard therapy in the treatment of HF by reducing sympathetic overdrive resulting in increased ejection fraction and survival [56]. However, β-AR blockers may also improve outcomes through its effects on energy substrate utilization and metabolic efficiency by using oxygen-sparing mechanisms [355,356]. As fatty acid oxidation in the heart utilizes more oxygen per unit of ATP generation, it is less efficient as compared to glucose oxidation [171,172,357]. Early studies studying the effects of β-AR blockage on general metabolism suggests that β-AR antagonists induce hypoglycemia and decrease the breakdown of glycogen [358,359,360,361]. Some clinicians are reluctant to prescribe β-AR blockers for HF because of negative inotropic and these general metabolic effects, including loss of glycemic control and insulin resistance [358,359,360,361]. However, the major effect of β-AR blockage in the heart is the suppression of lipolysis, decreased fatty acid uptake, and a reduction in myocardial fatty acid oxidation which may result in a compensatory increase in glucose oxidation and its resulting metabolic benefit in treating HF [362,363,364,365,366,367,368] (Figure 2). Furthermore, studies have shown that not all β-AR blockers are equal in their metabolic effects. Carvedilol, which is vasodilating (through α_1B_-AR blockage) in addition to its β-AR blockage, have metabolic effects that contribute to better outcomes in HF, compared with pure β-AR blockers [369,370,371,372,373,374,375]. 

Studies have also suggested that β-AR blockers may also increase cardiac mitochondrial respiration [376,377,378,379] and have anti-oxidative effects [380], along with regulating mitochondrial calcium levels and ADP uptake [381,382]. Mitochondrial abnormalities are associated with HF [383,384]. PPARs regulates many key metabolic oxidative processes and mitochondrial biogenesis and function [279,280,281]. As mentioned for α_1_-AR agonists coupling to PPARs, PPAR activators increase mitochondrial oxidative capacity in the heart [280] and is cardioprotective [282]. Isoproterenol-induced HF can be rescued though PPARα activation [284,287], suggesting that β_1_-AR stimulation decreases PPAR activation. Similarly, β_1_-AR autoantibodies induce a positive inotropy in the heart [385] but also induce HF [386,387], apoptosis [388] and inhibit the PPAR pathway [389,390]. However, the metabolic protective effects of a β_2_-AR agonist, higenamine, against mitochondrial and respiratory dysfunction is mediated though increased PPARα signals [391] (Figure 2). These results suggest that β_1_-AR selective blockers may be better at increasing mitochondrial function during HF than non-selective β-blockers.

Besides generating ATP, mitochondria are the main produces of ROS, generated during oxidative phosphorylation [392,393]. Mitochondrial-targeted antioxidants are gaining acceptance as an emerging therapy for HF [394]. Results from two small clinical studies suggest β-AR blockers can decrease fatty acid uptake and oxidation [366,368] (Figure 2), while increasing LV function in the absence of increased oxygen utilization which would limit ROS formation [394,395]. In patients with HF, long-term usage of β-AR blockers reduced the metabolic demand of oxygen on the heart and lowered oxidative stress and ROS damage [102,396,397,398,399,400,401,402]. Therefore, β-AR blockers would also limit ROS damage, an additional benefit as a metabolic therapeutic in HF.

The best described β-AR blocker and its effects on cardiac metabolism is carvedilol. Carvedilol is a β-AR antagonist and an α_1_-AR antagonist [403]. The α_1_-AR blocking ability of carvedilol has been downplayed in its cardiac protective potential as non-selective α_1_-AR blockers can increase the morbidity of heart failure [404]. However, carvedilol has a little higher selectivity for β_2_- versus β_1_-ARs [405,406] and has 10-fold higher affinity for the α_1B_-ARs versus α_1A_-ARs and binds with higher affinity at α_1_-ARs than at β_1_-ARs [407]. A possible explanation for carvedilol’s better outcomes in HF when compared to other β-AR blockers is its ability to selectivity block the α_1B_-AR. The α_1B_-AR, unlike the α_1A_-AR [1], mediates maladaptive effects on the heart [35,84,85,407] and does not mediate glucose metabolism in the heart [234,235,250], but still would produce vasodilation that has positive effects on heart function by decreasing heart rate and vascular resistance. Therefore, selective blockage of the α_1B_-AR by carvedilol would decrease its maladaptive effects in the heart and not disrupt the metabolic and other cardioprotective benefits of the α_1A_-AR.

In clinical studies, carvedilol has better cardioprotective metabolic properties than other β-blockers and may be a preferred treatment for HF. Carvedilol is better than metoprolol in its antioxidative properties [408,409,410,411] and has more favorable effects on glucose metabolism [412] and calcium load [413]. Carvedilol shifts the heart energy substrate usage from fatty acids to glucose oxidation [315,368] by decreasing free fatty acids, inhibiting fatty acid oxidation, resulting in increased glycolysis, oxidation and energy efficiency [364,395,414,415] (Figure 2). Mechanisms of cardioprotective effects of carvedilol are linked to oxidative metabolism and decreases in oxidative stress [102,327,363,414,415,416,417,418], which reduces the impairment of mitochondrial metabolism during HF [419]. Carvedilol can also improve insulin sensitivity and plasma lipid profile [419,420,421]. The net effect of these metabolic changes are favorable effects on glucose metabolism [412,421], by improving myocardial energy efficiency through increased carbohydrate utilization [395,422,423], similar to α_1A_-AR agonism. 

Carvedilol’s mechanism to shift metabolism to glucose utilization is also an AMPK-mediated mechanism, similar to α_1_-AR stimulation [424]. AMPK can increase fatty acid oxidation by increasing fatty acid uptake, increasing the expression of the fatty acid transporters, and by decreasing levels of malonyl-CoA, a potent inhibitor of carnitine palmitoyltransferase 1 [425,426] (Figure 2). β-ARs have been shown to regulate AMPK in the heart [427] and the decrease in responsiveness in β-ARs during heart failure and its associated pathological remodeling has been linked to β-AR’s associated loss of AMPK signals [428,429,430]. As β_1_-ARs are downregulated in HF, β-AR blockers may reactivate AMPK pathways to shift metabolism to glucose utilization. The β_3_-AR is also postulated to mediate its exercise-mediated cardioprotection through activation of AMPK signals [431]. 

## 6. Conclusions

While inotropic interventions for HF might improve contractile function for the short-term, chronic treatments have failed to improve outcomes. During the progression of HF there is a tremendous amount of metabolic inflexibility [273] caused by the progressive dysfunction of the metabolic pathways that generate ATP which starve the heart. Our current understanding of the failing heart suggests that optimizing energy substrate metabolism by inhibiting fatty acid oxidation while increasing glucose substrate availability in conjunction with its oxidation may provide a means to increase the efficiency of ATP production that is needed to maintain the high energy demands of cardiac function. Since severe HF also has diminished mitochondrial function, metabolic therapies that are multi-faceted and regulate key metabolic sensors may provide greatest benefit during all of the stages of HF. α_1_- and β-ARs regulate cardiac metabolism in opposition with α_1A_-AR agonism and β-AR blockage converging on increasing glucose availability and oxidation, suggesting that a dual action drug might provide the greatest therapeutic benefit. The ability of β-AR antagonists and α_1A_-AR agonists to regulate, optimize, and shift energy pathways as needed by the heart during failure are also predicted to lead to better outcomes.

## Figures and Tables

**Figure 1 ijms-22-05783-f001:**
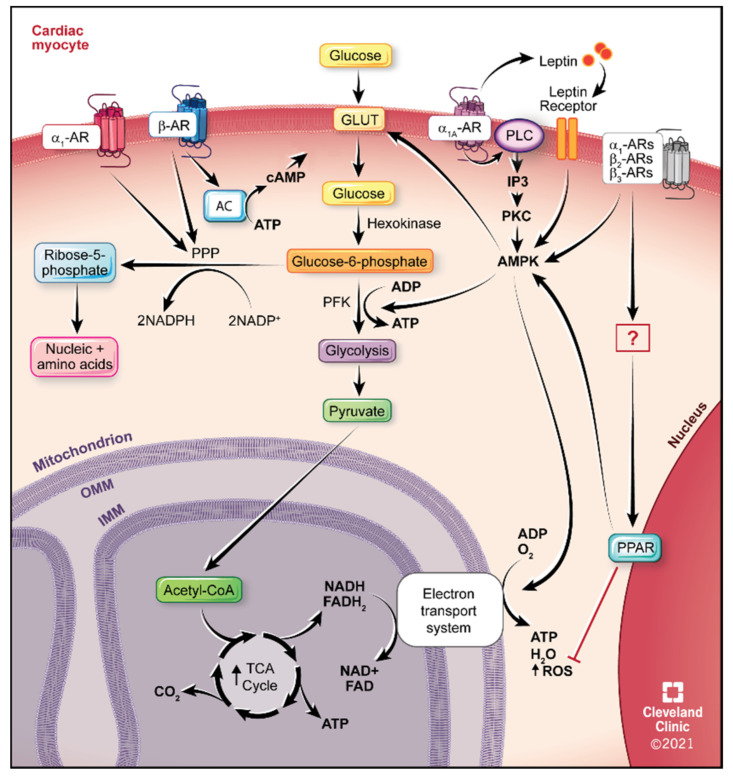
Adrenergic pathways that affect glucose utilization in the cardiac myocyte, showing glycolysis, glucose oxidation inside the mitochondrion, and the alternative pentose phosphate pathway (PPP). AC, adenylate cyclase; ADP, adenosine diphosphate; AMPK, AMP-activated protein kinase; AR, adrenergic receptor; ATP, adenosine triphosphate; CoA, coenzyme A; cAMP, cyclic adenosine monophosphate; FAD, flavin adenine dinucleotide; GLUT, Glucose transporter; IMM, inner mitochondrial membrane; IP3, inositol triphosphate; OMM, outer mitochondrial membrane; PFK, phosphofructokinase; PKC, protein kinase C; PLC, phospholipase C; PPAR, peroxisome proliferator-activated receptor; NAD, nicotinamide adenine dinucleotide; NADP, nicotinamide adenine dinucleotide phosphate; ROS, reactive oxygen species; TCA, tricarboxylic acid cycle.

**Figure 2 ijms-22-05783-f002:**
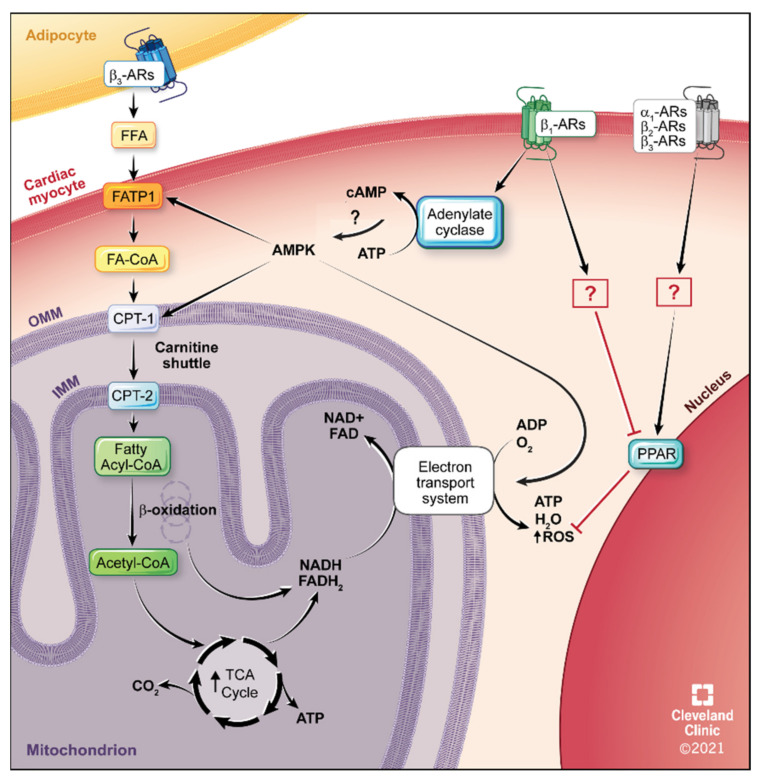
Adrenergic Receptor Regulation of Fatty Acid Metabolism in the Cardiac Myocyte. AC, adenylate cyclase; ADP, adenosine diphosphate; AMPK, AMP-activated protein kinase; AR, adrenergic receptor; ATP, adenosine triphosphate; CoA, coenzyme A; cAMP, cyclic adenosine monophosphate; CPT, carnitine palmitoyltransferase; FAD, flavin adenine dinucleotide; FATP1, Fatty acid transport protein 1; FFA, free fatty acid; IMM, inner mitochondrial membrane; NAD, nicotinamide adenine dinucleotide; OMM, outer mitochondrial membrane; PPAR, peroxisome proliferator-activated receptor; ROS, reactive oxygen species; TCA, tricarboxylic acid cycle.

**Table 1 ijms-22-05783-t001:** Properties of the AR Subtypes in the Cardiovascular System.

Subtype	Signal Transduction	Tissue Distribution	Physiological Function
α_1A_	G_q_/G_11_/PLC/PKC/ DAG/IP3/Ca^+2^	Cardiac myocyte Vascular smooth muscle	Positive inotropy, chronotropy, cardiac hypertrophy, contraction smooth muscle, blood pressure
α_1B_	G_q_/G_11_/PLC/PKC/ DAG/IP3/Ca^+2^	Cardiac myocyte Vascular smooth muscle	Negative inotropy, cardiac hypertrophy, contraction smooth muscle, blood pressure
α_1D_	G_q_/G_11_/PLC/PKC/ DAG/IP3/Ca^+2^	Coronary arteries Vascular smooth muscle	Contraction smooth muscle, blood pressure
α_2A_ α_2B_ α_2C_	Gi/inhibit AC/ cAMP/PKA	Not in any cardiac tissue Vascular smooth muscle	NE release- Sympathetic nerve endings
β_1_	Gs/AC/cAMP/PKA Ca^+2^ channel	Cardiac myocyte	Positive inotropy, chronotropy, cardiac hypertrophy
β_2_	Gs/AC/cAMP/PKA Ca^+2^ channel Gi/inhibit AC/ cAMP/PKA	Cardiac myocyte Vascular smooth muscle	Cardiac hypertrophy Relaxation smooth muscle
β_3_	Gs/Gi/AC/cAMP/PKA NO	Cardiac myocyte	Negative inotropy

AC, adenylate cyclase; Ca^+2^, calcium; cAMP, cyclic adenosine monophosphate; DAG, diacylglycerol; IP3, inositol triphosphate; NE, norepinephrine; NO, nitric oxide; PKA, protein kinase A; PLC, phospholipase C.

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
