# Peer review of "Targeting Adrenergic Receptors in Metabolic Therapies for Heart Failure"

_ijms, 2021, doi:10.3390/ijms22115783_

Round 1

Reviewer 1 Report

The Author reviews the current knowledge about the adrenergic receptors in the light of their role in the regulation of cardiomyocyte metabolism and their pharmacological targeting in heart failure therapy. I appreciated the section on cardiomyocyte metabolism in the normal and pathological conditions that precedes the discussion of the role of the adrenergic receptors in metabolism regulation.

I would like to suggest adding  1-2 sentences at the end of introduction about the scope and content of this review. In its present form, there is nothing that invites or motivates the reader to look further.

In the section 5, there are two subsections for alpha1 receptors dedicated, first, to their metabolic effects and, second, to their therapeutic potential, followed by only one very long subsection for beta receptors. I suggest it should be similarly divided in two parts.

There are some awkward, even if still perfectly comprehensive, phrases:

 "[...] can regulate different physiologies" (could be "physiological processes");

"The myocyte contains a mixture of [receptors]" (could be "expresses");

"The role of the beta3-AR is still controversial in its role in HF".

Author Response

Comment 1: The Author reviews the current knowledge about the adrenergic receptors in the light of their role in the regulation of cardiomyocyte metabolism and their pharmacological targeting in heart failure therapy. I appreciated the section on cardiomyocyte metabolism in the normal and pathological conditions that precedes the discussion of the role of the adrenergic receptors in metabolism regulation.

Response: Thank you for the kind comments about our review.  

Comment 2: I would like to suggest adding  1-2 sentences at the end of introduction about the scope and content of this review. In its present form, there is nothing that invites or motivates the reader to look further.

Response: We have now done so. This is now seen in lines 81-85 in the revised manuscript.

Comment 3: In the section 5, there are two subsections for alpha1 receptors dedicated, first, to their metabolic effects and, second, to their therapeutic potential, followed by only one very long subsection for beta receptors. I suggest it should be similarly divided in two parts.

Response: We have now done so. 

Comment 4: There are some awkward, even if still perfectly comprehensive, phrases:

 "[...] can regulate different physiologies" (could be "physiological processes");

"The myocyte contains a mixture of [receptors]" (could be "expresses");

"The role of the beta3-AR is still controversial in its role in HF".

Response: We have revised all these sentences as suggested.

Reviewer 2 Report

This review was interesting for me, summarizes knowledge about the use of adrenergic ligands for heart failure therapy from the perspective of their influence on metabolism. I only have a few comments on the formal aspect of the document. Some abbreviations are incorrectly introduced (eg PPARs are used before explanation or typo - PFC for PFK). In the list of literature, alpha and beta are indicated by another symbol.

Author Response

Comments: This review was interesting for me, summarizes knowledge about the use of adrenergic ligands for heart failure therapy from the perspective of their influence on metabolism. I only have a few comments on the formal aspect of the document. Some abbreviations are incorrectly introduced (eg PPARs are used before explanation or typo - PFC for PFK). In the list of literature, alpha and beta are indicated by another symbol.

Response: We are pleased you thought the review was interesting. We have now gone through the text and references to correct abbreviations and symbols.